# Application of explainable ensemble artificial intelligence model to categorization of hemodialysis-patient and treatment using nationwide-real-world data in Japan

Eiichiro Kanda[1]*, Bogdan I. Epureanu[2], Taiji Adachi[3], Yuki Tsuruta[4], Kan Kikuchi[5], Naoki Kashihara[6], Masanori Abe[7], Ikuto Masakane[8], Kosaku Nitta[9]

1 Medical Science, Kawasaki Medical School, Kurashiki, Okayama, Japan, 2 College of Engineering, University of Michigan, Ann Arbor, Michigan, United States of America, 3 Institute for Frontier Life and Medical Sciences, Kyoto University, Sakyo, Kyoto, Japan, 4 Tsuruta Itabashi Clinic, Itabashi, Tokyo, Japan, 5 Shimoochiai Clinic, Shinjuku, Tokyo, Japan, 6 Department of Nephrology and Hypertension, Kawasaki Medical School, Kurashiki, Okayama, Japan, 7 Division of Nephrology, Hypertension and Endocrinology, Department of Internal Medicine, Nihon University School of Medicine, Itabashi, Tokyo, Japan, 8 Department of Nephrology, Yabuki Hospital, Yamagata, Yamagata, Japan, 9 Department of Nephrology, Tokyo Women's Medical University, Shinjuku, Tokyo, Japan

☯ These authors contributed equally to this work.
* kms.cds.kanda@gmail.com

## Abstract

### Background

Although dialysis patients are at a high risk of death, it is difficult for medical practitioners to simultaneously evaluate many inter-related risk factors. In this study, we evaluated the characteristics of hemodialysis patients using machine learning model, and its usefulness for screening hemodialysis patients at a high risk of one-year death using the nation-wide database of the Japanese Society for Dialysis Therapy.

### Materials and methods

The patients were separated into two datasets (n = 39,930, 39,930, respectively). We categorized hemodialysis patients in Japan into new clusters generated by the K-means clustering method using the development dataset. The association between a cluster and the risk of death was evaluated using multivariate Cox proportional hazards models. Then, we developed an ensemble model composed of the clusters and support vector machine models in the model development phase, and compared the accuracy of the prediction of mortality between the machine learning models in the model validation phase.

### Results

Average age of the subjects was 65.7±12.2 years; 32.7% had diabetes mellitus. The five clusters clearly distinguished the groups on the basis of their characteristics: Cluster 1, young male, and chronic glomerulonephritis; Cluster 2, female, and chronic glomerulonephritis; Cluster 3, diabetes mellitus; Cluster 4, elderly and nephrosclerosis; Cluster 5, elderly

**Data Availability Statement:** Data cannot be made publicly available by the authors, as they are owned by the Japanese Society for Dialysis Therapy.

Interested readers may request the data at the following URL: http://www.jsdt.or.jp/jsdt/1761. html.

**Funding:** This work was supported by Japan Society for the Promotion of Science (KAKENHI Grant Number JP 19K08740) to EK. The funder had no role in study design, data collection and analysis, decision to publish, or preparation of the manuscript.

**Competing interests:** The authors have declared that no competing interests exist.

and protein energy wasting. These clusters were associated with the risk of death; Cluster 5 compared with Cluster 1, hazard ratio 8.86 (95% CI 7.68, 10.21). The accuracy of the ensemble model for the prediction of 1-year death was 0.948 and higher than those of logistic regression model (0.938), support vector machine model (0.937), and deep learning model (0.936).

## Conclusions

The clusters clearly categorized patient on their characteristics, and reflected their prognosis. Our real-world-data-based machine learning system is applicable to identifying high-risk hemodialysis patients in clinical settings, and has a strong potential to guide treatments and improve their prognosis.

## Introduction

The mortality rates of dialysis patients are very high and the number of prevalent end-stage-kidney disease (ESKD) patients has been increasing in the USA and Japan [1, 2]. To improve their prognosis, early identification of patients at a high risk of death, and interventional treatments of their conditions are necessary.

Various risk factors for death in dialysis patients have been identified [3–5]. These risk factors are associated with each other forming a complex network which should be simultaneously taken into account and controlled [6–8]. The Dialysis Outcomes and Practice Patterns Study (DOPPS) has defined a survival index to predict the hemodialysis patients' risk of death using logistic regression models [6]. We also have developed a nutritional risk index (NRI) for hemodialysis patients using Cox proportional hazards models [8]. However, these indices make some statistical assumptions which limit their application; they also take time to calculate, which is inconvenient when dealing with many patients in clinical settings. The development of a new automatic system is needed to help manage various risk factors simultaneously, and to improve the prognosis of a large number of patients.

Artificial intelligence (AI) methods hold great promise for decision-making in complex systems including those used in medicine for diagnosis and prediction [9, 10]. Although AI is useful to accurately diagnose patients at a high risk of death, only a few studies on the prediction of ESKD patients' prognosis have been carried out [11–13]. Difficulties constructing AI algorithms for clinical use have been pointed out, such as the scarce availability of reliable and large data sets for AI algorithm construction, the lack of transparency of conventional AI algorithms, the difficult integration of AI algorithms into complex existing clinical work flow, and the cumbersome compliance with regulatory medical frameworks [14]. Overcoming some or all of these difficulties is required to create a new AI-based system for ESKD patients.

Therefore, in this study, we aim to establish an implementable AI system for screening hemodialysis patients at a high risk of death and for predicting their prognosis on the basis of real-world data from the Japanese Society for Dialysis Therapy (JSDT) Renal Data Registry (JRDR). JRDR is a nationwide-data registry and includes 98.8% of ESKD patients in Japan [1]. To provide transparency and accuracy of AI predictions, an ensemble model composed of the K-means method and a support vector machine (SVM) was developed. Then, the performance of the proposed model was compared with that of a SVM-alone model, a deep learning model, and a multivariate logistic regression model. Moreover, considering their usage and applicability to clinical settings, we developed a new total-care system for treating hemodialysis patients at a high risk of death.

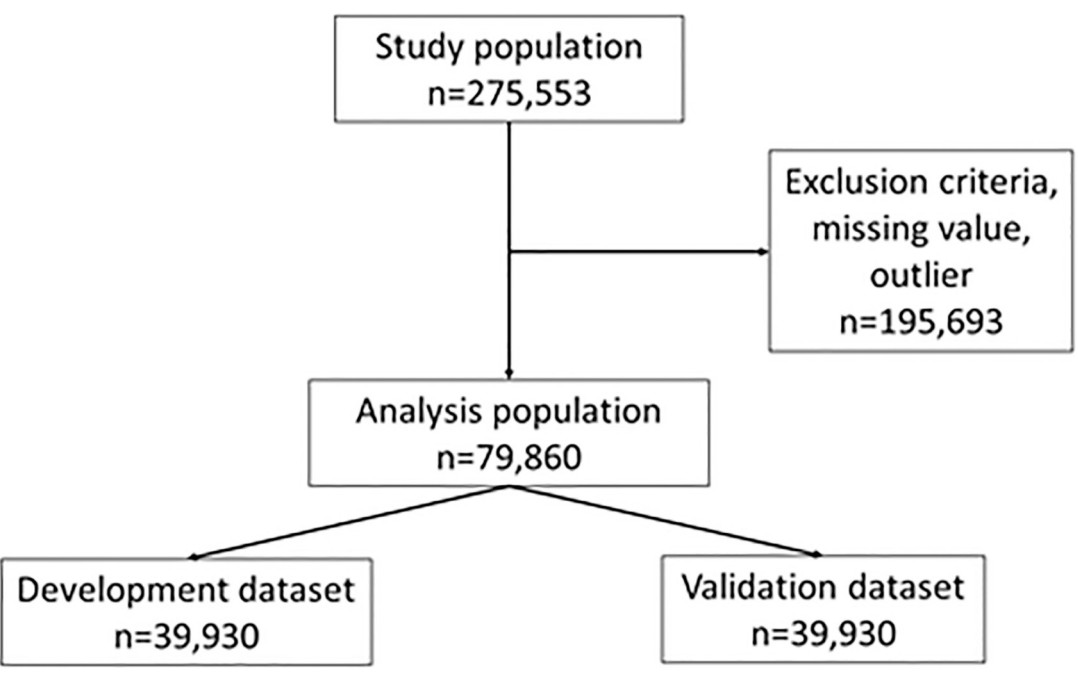

**Fig 1. Randomization of study population and datasets.**

## Materials and methods

### Dataset

This is a prospective cohort study of maintenance hemodialysis patients using JRDR data. JSDT has been conducting annual surveys of dialysis facilities in Japan since 1968. The JRDR data from 2008 to 2013 were used in this study. This study was approved by the ethics committee of JSDT and was exempt from the need to obtain informed consent from participants (JSDT No. 33). The data were analyzed anonymously. The study was performed in accordance with the relevant guidelines and the Declaration of Helsinki of 1975 as revised in 1983.

The subjects of this study were the 275,553 patients (Fig 1). The exclusion criteria were as follows: patients younger than twenty years; patients on hemodiafiltration, hemofiltration, or peritoneal dialysis; patients with missing values or outlier values of laboratory data; patients who had a limb amputated; and patients with a hemodialysis vintage of less than one year. Thus, 79,860 patients were included in the analysis. The included subjects were randomly classified into two groups to obtain a dataset for the development of the machine learning algorithms (development dataset, 39,930) and a dataset for the validation of the algorithms (validation dataset, 39,930).

The endpoints were all-cause death within one and five years. The data of the baseline characteristics were as follows: age; gender; diabetes mellitus (DM), chronic glomerulonephritis (CGN), or nephrosclerosis as a cause of ESKD; history of cardiovascular disease (CVD); body mass index (BMI); serum albumin, sodium, potassium, calcium, phosphorus, creatinine, total cholesterol, and C-reactive protein (CRP) levels; hemoglobin level; normalized protein catabolic rate (nPCR); vintage; Kt/V; and ultrafiltration ratio. The laboratory data were measured before hemodialysis, and BMI was calculated using the weight measured after hemodialysis. Nutritional status was evaluated using NRI for hemodialysis patients [8]. NRI is a nutritional screening index used to predict hemodialysis patients' prognosis, and was developed by JSDT.

It is calculated as follows:

$$\text{Risk score} = \text{low BMI} + \text{low serum albumin level} + \text{abnormal serum total cholesterol level} + \text{low serum creatinine level} \quad (1)$$

where the above parameters are defined as followed: low BMI ($<20\text{kg/m}^2$), yes = 3, no = 0; low serum albumin level (age $<65$, $<3.7\text{g/dL}$; age $\geq65$, $<3.5\text{g/dL}$), yes = 4, no = 0; abnormal serum total cholesterol level, low($<130\text{mg/dL}$) = 1, high ($\geq220\text{mg/dL}$) = 2, no = 0; low serum creatinine level (age $<65$, male $<11.6$, female $<9.7\text{mg/dL}$; age $\geq65$, male $<9.6$, female $<8.0\text{mg/dL}$), yes = 4, no = 0. Risk of 1-year death: low risk, risk score = 0 to 7; medium risk, score = 8 to 10; high risk, score = 11 and higher.

## Statistical analyses

Normally distributed variables are presented as mean±standard deviation; otherwise, the median and interquartile ranges are presented. Highly skewed variables were transformed with the natural logarithm function prior to use in models [ln(vintage), ln(CRP)]. Intergroup comparisons of parameters were performed using the chi-square test, t-test, Mann-Whitney U test, one-way analysis of variance, and the Kruskal-Wallis test as appropriate. These analyses were conducted using SAS version 9.4 (SAS, Inc., NC, USA), R version 3.4.1 (R project for Statistical Computing, Vienna, Austria), and Python version 3.7.4 (Python Software Foundation, DE, USA). Statistical significance was defined as $p < 0.05$.

## Development of machine learning models

The variables of the baseline characteristics were Z-score-normalized and used for the following modeling.

**K-means method model.** Step 1: Patients were grouped into clusters from 2 to 10 on the basis of their baseline characteristics by the K-means method using the basis of the development dataset. Patients with similar characteristics were grouped in one cluster, and the patients in other clusters showed dissimilar characteristics. First, patients were randomly selected as initial cluster centers. Next, each patient was assigned to one cluster on the basis of the closeness of their characteristics to the cluster center. The mean of samples in a cluster was calculated as the new cluster center, $\mu$. These steps were repeated until the final stable clustering results were obtained. The similarity between a patient $x$ and a center $\mu$ in a cluster was evaluated using the Euclidean distance in an m-dimensional space, $dist(x,\mu)$:

$$dist(x, \mu)^2 = \sum_{j=1}^{m}(x_j - \mu_j)^2 = \|x - \mu\|^2 \quad (2)$$

where $j$ is the $j^{\text{th}}$ variable of the baseline characteristics, $m$ is the number of variables of the baseline characteristics; in this study, $m = 20$.

Step 2: To evaluate the clustering, the within-cluster sum of squared errors (SSEs), namely, distortion $J$, was measured:

$$J = \sum_{i=1}^{n}\sum_{j=1}^{k}r_{ij}\|x_i - \mu_j\|^2 \quad (3)$$

where $\mu_j$ is the center for cluster $j$, if $x_i$ is in cluster $j$, $r_{ij} = 1$, else $r_{ij} = 0$, $k$ is the number of clusters, and $n$ is the number of patients.

To use a gradient-based optimizer for $J$, Eq (3) is partially differentiated by $\mu_j$ to obtain:

$$\frac{\partial J}{\partial \mu_j} = 2\sum_{i=1}^{n}r_{ij}(x_i - \mu_j) = 0 \quad (4)$$

$$\mu_j = \frac{\sum r_{ij} x_i}{\sum r_{ij}} \tag{5}$$

The elbow method was used to identify the number of clusters where the within-cluster SSE decreased rapidly.

Next, to evaluate whether the clusters could discriminate the patients on the basis of their risks of the endpoints, the survival probabilities of the clusters were evaluated using Kaplan-Meier survival curves. The clusters were indicated by numbers on the basis of the risks, and Cox proportional hazards models were evaluated to compare the risk of an endpoint between clusters. The Cox proportional hazards models were developed including only the cluster used as a categorical variable because the K-means method can be considered as a function which was composed of variables of the baseline characteristics:

$$\text{Cluster } i = f(x_1, x_2, \ldots, x_m) \tag{6}$$

Hazard ratio results (HRs) with 95% confidence interval (CI) are presented here.

Step 3: The patients in the validation dataset were grouped into clusters using the K-means method trained using the development dataset. Then, the relationship between the clusters and the risk of the endpoints were evaluated using Kaplan-Meier survival curves, and Cox proportional hazards models. Considering the results, the optimal number of clusters, $k$, was determined, and the differences in characteristics between the clusters were statistically evaluated.

**Multivariate logistic regression model.**   To predict the probabilities of the endpoints, multivariate logistic regression models (LRMs) including all variables of the baseline characteristics were developed using the development dataset as follows:

$$\log\left(\frac{p}{1-p}\right) = \alpha + \sum_{i=1}^{m} \beta_i x_i \tag{7}$$

$$p = \frac{1}{1 + \exp(-\sum_{i=1}^{m} \beta_i x_i)} \tag{8}$$

where $x_i$ is the $i$th variable of the baseline characteristics, and $\beta_i$ is the parameter estimate for the same variable.

When $p$ was estimated to be more than 0.5, a patient's death was predicted. Then, using the validation dataset, we evaluated the accuracy of the prediction using the LRMs.

**Support vector machine models.**   SVM models were used to predict the endpoints. SVM models with a Gaussian radial basis function kernel included all of the variables of the baseline characteristics. In the development of each SVM model, classification was examined on the basis of the three-fold cross validation method, and the accuracy of the prediction was estimated by taking the three results. Then, the final SVM models were developed. Using the validation dataset, we evaluated the accuracy of the prediction of the endpoints using the SVM models developed.

**Ensemble model.**   Using the development dataset, we grouped the patients into the $k$ clusters previously determined by the K-means method (Fig 2). Each SVM model including all of the variables of the baseline characteristics for each cluster was trained to predict the risk of

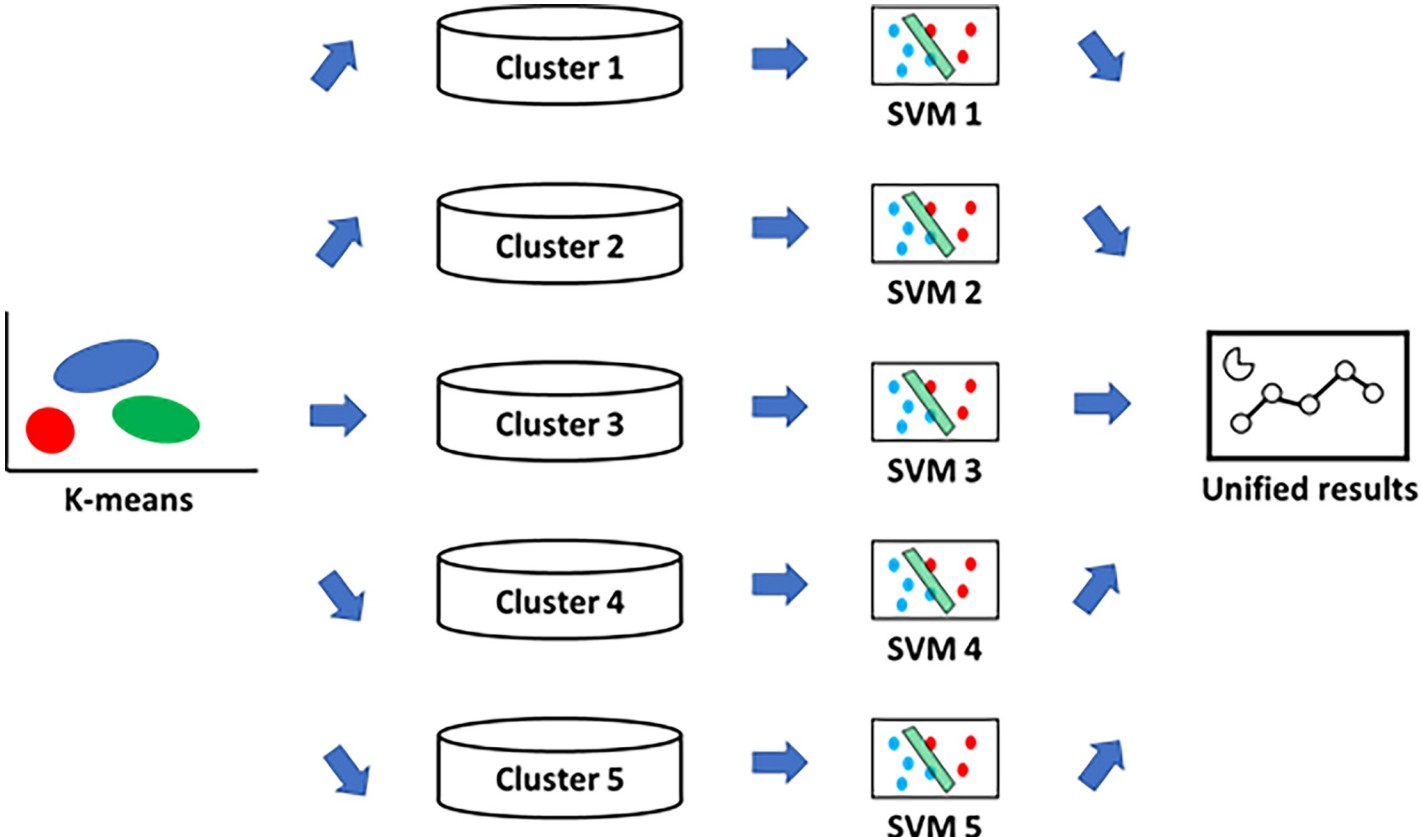

**Fig 2. Structure of ensemble model.** The patients were grouped into five clusters using the K-means method. Each cluster was analyzed using each SVM model. The results of analysis using SVM models were unified. Abbreviation: SVM, support vector machine.

the endpoints. And $k$ SVM models, $\Phi(Cluster\ i)$, were developed.

$$\Phi(x) = (\varphi(Cluster\ 1), \varphi(Cluster\ 2), \ldots, \varphi(Cluster\ k)) \tag{9}$$

where $\varphi(Cluster\ i)$ is a SMV model for Cluster i.

Then, the patients in the validation dataset were grouped into $k$ clusters, and the trained SVM models were applied to the corresponding clusters. The results of the prediction of endpoints were unified.

**Deep learning models.** Deep learning models were developed to predict death at 1-year and 5-years of dialysis (1-year and 5-year deaths, respectively). The numbers of layers and hyperparameters were optimized on the basis of the accuracy to predict the endpoints and to prevent overfitting (Figs 3 and 4). In the development of each deep learning model, two-thirds of the development dataset was used as the training dataset and the remaining one-third was used as the test dataset. Then, using the validation dataset, we evaluated the accuracy of the prediction of the endpoints using the deep learning models.

**Evaluation of model performance.** The performance of the models developed for the binary diagnosis decision (death or no death) in terms of accuracy, sensitivity, and specificity was evaluated using the validation dataset. Accuracy is calculated as follows:

$$accuracy = sensitivity \times risk\ of\ death + specificity \times (1 - risk\ of\ death) \tag{10}$$

$$\text{risk of death} = \frac{number \ of \ death}{total \ number \ of \ the \ patients} \tag{11}$$

Because of this method chosen to calculate accuracy, the value of accuracy changes depending on the number of endpoints (the risk of death). Here, given that sensitivity and specificity were constant, we simulated accuracy at various risks of death from 0.05 to 0.65, and compared the accuracies of the models.

## Results

### Baseline characteristics

The baseline characteristics including biochemical data are shown in Table 1. No statistically significant differences in the baseline characteristics between the development and validation datasets were observed. Machine learning models were constructed (Fig 5).

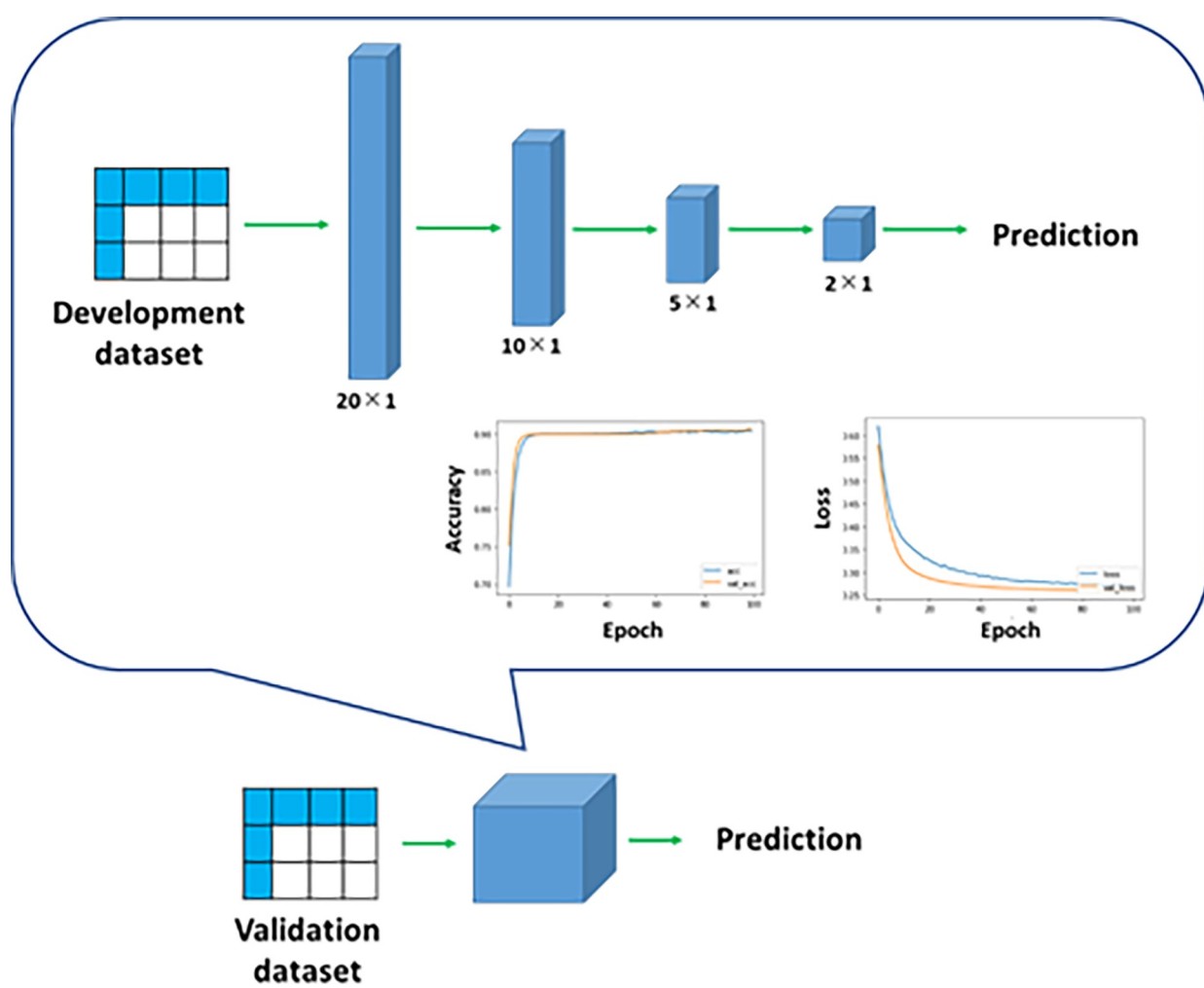

**Fig 3. Deep learning model for prediction of 1-year death.** The models for the prediction of 1-year and 5-year deaths were composed of multiple layers: 1 input layer; 2 or 4 hidden layers, respectively; and 1 output layer. Data of a patient were treated as 1 vector of 20 dimensions. Through the hidden layers, the patient's characteristics were extracted. The dropout rate of each hidden layer was determined appropriately. Adam was used as a learning rate optimization algorithm. ReLUs were used as the activation function of hidden layers, and the logistic activation function was used in the output layer. The performance of a deep learning model was evaluated in terms of accuracy and loss function. The trained model was applied to the validation dataset.

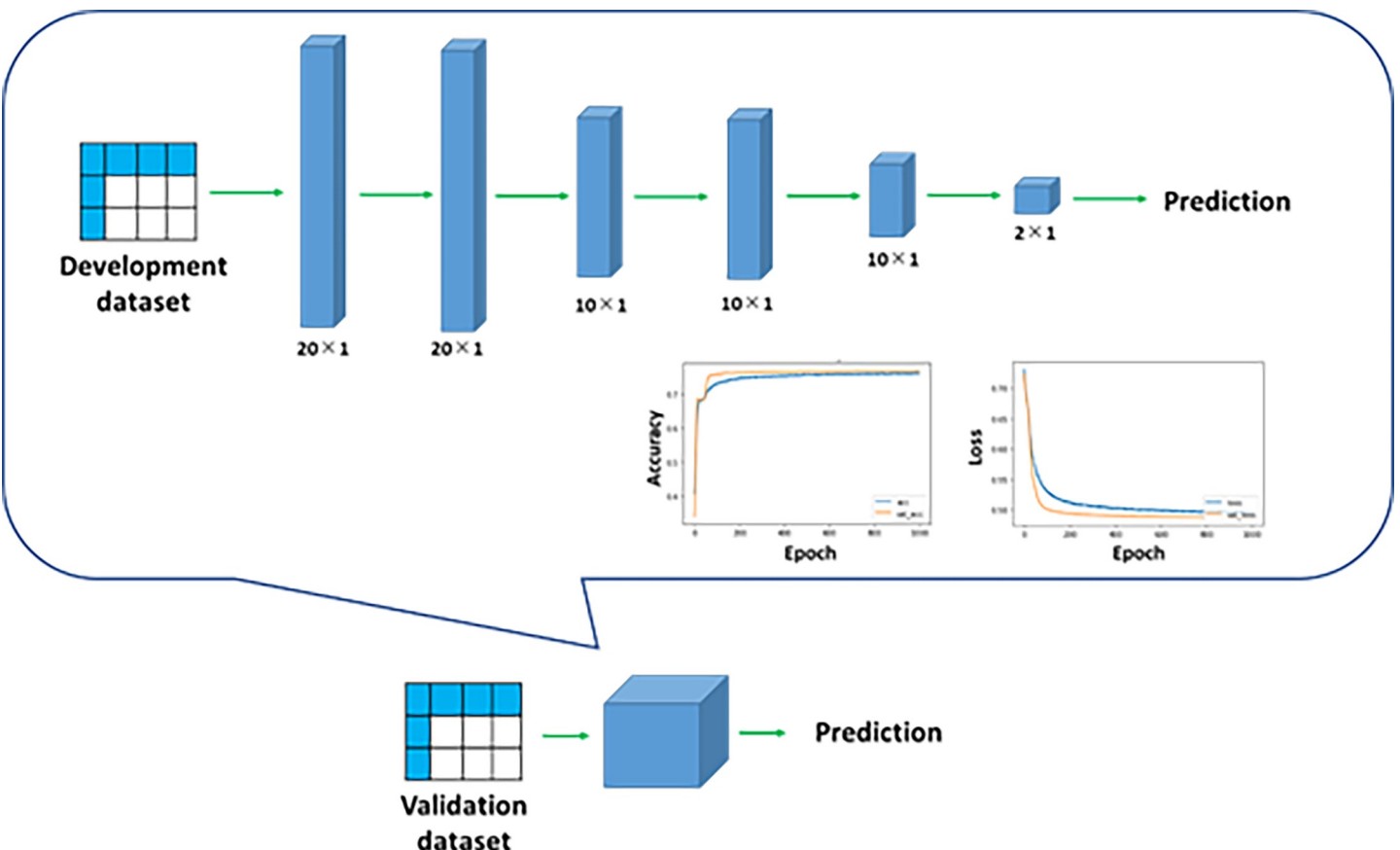

**Fig 4. Deep learning model for prediction of 5-year death.** The deep learning model for the prediction of 5-year death was developed similarly to the model for 1-year death. The trained model was applied to the validation dataset.

## K-means method models

The K-means method was conducted, and the models with 2 to 10 clusters were developed. The elbow method showed decreasing in within-cluster SSE with increasing numbers of clusters (Fig 6). Five and six clusters were chosen as candidate numbers of clusters.

The Kaplan-Meier survival curves showed the relationship between the numbers of clusters and the risk of death (Figs 7 and 8). The five-cluster model clearly distinguished the patients on the basis of the risk of 1-year and 5-year deaths both in the development and validation datasets (Figs 7A, 7B, 8A and 8B); (Tables 2A and 3A). Cluster 5 showed the highest risks of 1-year and 5-year deaths.

In contrast, the six-cluster model showed that the rank of the clusters based on the risk of 1-year death in the development dataset was different from the rank in the validation dataset (Table 2B). Although the risk of 1-year death of Cluster 2 (HR, 1.87) was lower than that of Cluster 3 (HR, 2.55) in the development dataset, the risk of Cluster 2 (HR, 1.58) was higher than that of Cluster 3 (HR, 1.51). Moreover, Cluster 6 showed the highest risk of 5-year death in the development dataset (Table 3B). However, in the validation dataset, the risk of Cluster 5 was very close to that of Cluster 6 (Table 3B); (Fig 8C and 8D), which suggests that the six-cluster model might be unreliable in reflecting the patients' prognosis depending on the patient data. Therefore, considering the stability of the accuracy of the five-cluster model in reflecting the patients' prognosis, $k = 5$ was considered appropriate for the model, and the five-cluster model was hereafter adopted.

**Table 1. Baseline characteristics.**

| | All | Development dataset | Validation dataset | p |
|---|---|---|---|---|
| N | 79,860 | 39,930 | 39,930 | |
| Age (years) | 65.7 ± 12.2 | 65.7 ± 12.2 | 65.6 ± 12.2 | 0.28 |
| Male (%) | 49084 (61.5) | 24537 (61.5) | 24547 (61.5) | 0.95 |
| DM (%) | 26154 (32.7) | 13200 (33.1) | 12954 (32.4) | 0.065 |
| CGN (%) | 31758 (39.8) | 15829 (39.6) | 15935 (39.9) | 0.45 |
| Nephrosclerosis (%) | 5343 (6.7) | 2673 (6.7) | 2670 (6.7) | 0.98 |
| CVD (%) | 14998 (18.8) | 7530 (18.9) | 7468 (18.7) | 0.58 |
| BMI (kg/m$^2$) | 21.2 ± 3.4 | 21.2 ± 3.4 | 21.2 ± 3.4 | 0.24 |
| Albumin (g/dL) | 3.7 ± 0.4 | 3.7 ± 0.4 | 3.7 ± 0.4 | 0.99 |
| Sodium (mEq/L) | 138.9 ± 3.2 | 138.9 ± 3.2 | 138.9 ± 3.2 | 0.44 |
| Potassium (mEq/L) | 5 ± 0.8 | 5 ± 0.8 | 5 ± 0.8 | 0.78 |
| Calcium (mg/dL) | 9.3 ± 0.8 | 9.3 ± 0.8 | 9.3 ± 0.8 | 0.65 |
| Phosphorus (mg/dL) | 5.3 ± 1.4 | 5.3 ± 1.4 | 5.3 ± 1.4 | 0.88 |
| Creatinine (mg/dL) | 10.7 ± 2.8 | 10.7 ± 2.8 | 10.7 ± 2.8 | 0.79 |
| Total cholesterol (mg/dL) | 153.7 ± 34.5 | 153.4 ± 34.4 | 154 ± 34.6 | 0.052 |
| CRP (mg/dL) | 0.49 ± 1.42 0.11 (0.05, 0.34) | 0.5 ± 1.41 0.11 (0.05, 0.34) | 0.49 ± 1.43 0.11 (0.05, 0.34) | 0.41 |
| Hemoglobin (g/dL) | 10.4 ± 1.2 | 10.4 ± 1.2 | 10.4 ± 1.2 | 0.92 |
| nPCR (g/kg/day) | 0.88 ± 0.17 | 0.88 ± 0.17 | 0.88 ± 0.17 | 0.91 |
| Vintage (years) | 8.3 ± 6.7 6.2 (3.3, 11.1) | 8.2 ± 6.7 6.2 (3.3, 11.1) | 8.3 ± 6.7 6.3 (3.3, 11.1) | 0.13 |
| Kt/V | 1.4 ± 0.3 | 1.4 ± 0.3 | 1.4 ± 0.3 | 0.68 |
| Ultrafiltration (%) | 4.4 ± 1.8 | 4.4 ± 1.8 | 4.4 ± 1.8 | 0.062 |
| 1-year death (%) | 5234 (6.6) | 2649 (6.7) | 2585 (6.5) | 0.37 |
| 5-year death (%) | 25410 (31.8) | 12709 (31.8) | 12701 (31.8) | 0.96 |

Variables are expressed as mean±standard deviation. Vintage and CRP are also shown as median and interquartile range. Intergroup comparisons of parameters were performed using the chi-square test, t-test, and the Mann-Whitney U test as appropriate.

Abbreviations: DM, diabetes mellitus as a cause of end-stage renal disease; CGN, chronic glomerulonephritis; CVD, cardiovascular disease; BMI, body mass index; CRP, C-reactive protein; nPCR, normalized protein catabolic rate.

## Difference in the characteristics among five clusters

The five-cluster model could cluster the patients on the basis of their characteristics (Table 4). The mean ages of Clusters 4 and 5 were older than those of other groups. A gender difference was observed; most of the patients in Cluster 1 were males (94.2%), and those in Cluster 2 were females (92.3%). There were also significant differences in the causes of ESKD between the groups as follows: Clusters 1 and 2, CGN (74.6%, and 60.3%, respectively); Cluster 3, DM (93.3%); Cluster 4, nephrosclerosis (100%). In Cluster 5, the numbers of patients with DM and CGN were almost the same as the mean numbers in the study population (Tables 1 and 4). Moreover, the numbers of patients who had a history of CVD were larger in Clusters 3 to 5 than in Clusters 1 and 2.

There were significant differences in the laboratory data among the clusters. Serum albumin and potassium levels gradually decreased with increasing in clusters number. The serum phosphorus, and creatinine levels; and nPCR in Cluster 5 were lower than those in the other groups. The number of patients with high and medium risk of NRIs were larger in Clusters 4 and 5 than in the other clusters. The CRP levels in Clusters 4 and 5 were higher than those in other groups.

The risk of all-cause death in Cluster 5 was higher than those in the other groups (Table 5). The trends similar to all-cause death were observed in the risks of CVD- and infection-caused deaths. The details of 5-year death were as follows. The proportions of CVD-caused death in

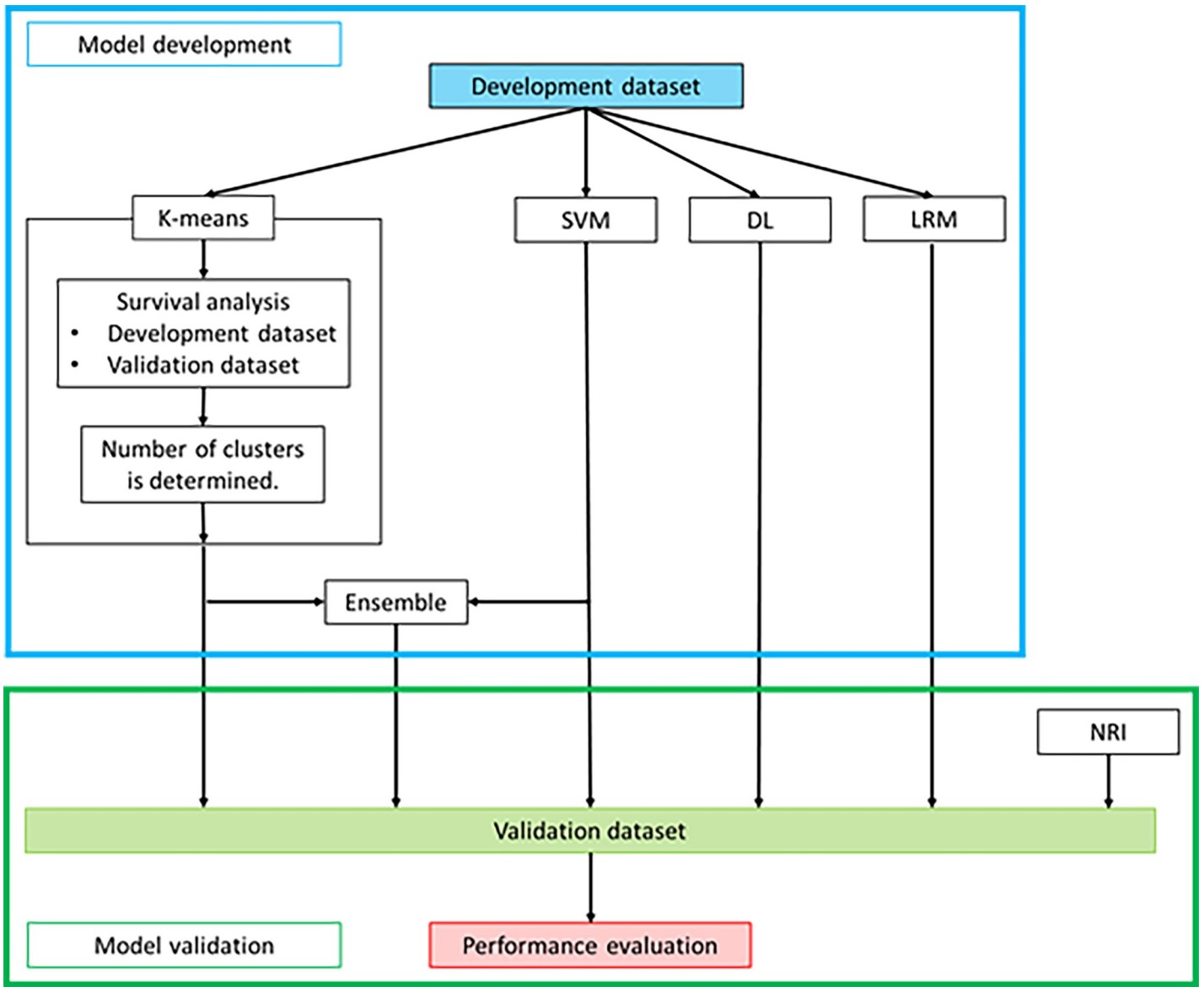

**Fig 5. Development of machine learning models and comparison of their performance.** This study consisted of the model development phase and model validation phase. In the model development phase, the K-means method model, SVM model, ensemble model, DL model, and LRM were developed. In the model validation phase, the performances of the models and NRI were compared. Abbreviations: SVM, support vector machine; DL, deep learning; LRM, logistic regression model; NRI, nutritional risk index.

5-year death were Cluster 1, 27.3%; Cluster 2, 27.4%; Cluster 3, 26.5%; Cluster 4, 27.4%; and Cluster 5 29.8% ($p < 0.0001$). And those of infection-caused death were Cluster 1, 5.3%; Cluster 2, 4.6%; Cluster 3, 3.9%; Cluster 4, 6.6%; and Cluster 5, 8.0% ($p < 0.0001$).

## Performance of models to predict death

The five-cluster model had four cutoff points. The accuracies of predicting death on the basis of these cutoff points were compared with those of the LRM, SVM model, ensemble model, deep learning model, and the high-risk group of NRI (Fig 9). The accuracies of predicting 1-year and 5-year deaths using LRM (0.938, 0.759), SVM model (0.937, 0.758), ensemble

## Within-cluster SSE

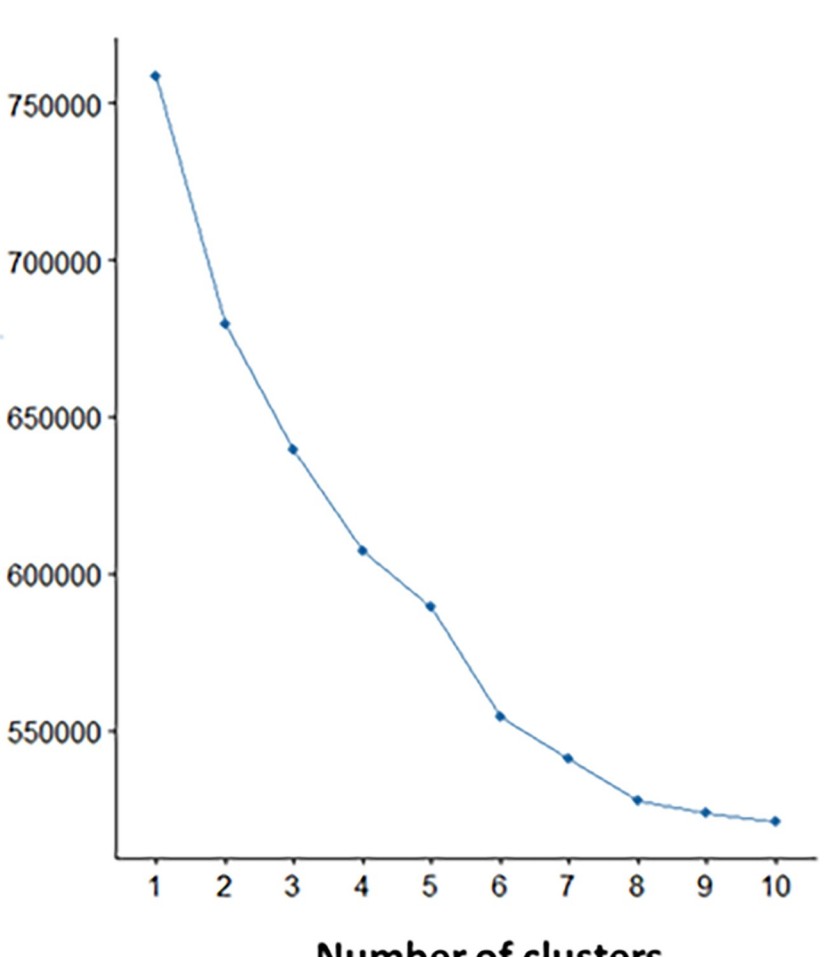

**Fig 6. Elbow method.** Elbow method shows the relationship between within-cluster SSE and number of clusters. Abbreviation: within-cluster SSE, within-cluster sum of squared errors.

model (0.948, 0.755), and deep learning model (0.936, 0.756) were higher than those using the five-cluster model with Clusters 4 as the cutoff point (0.809, 0.716). The ensemble model showed the highest accuracy to predict 1-year death. The accuracies of predicting 5-year death using LRM, SVM model, ensemble model, and deep learning model were almost the same, and were higher than those using the five-cluster models and NRI.

The estimated accuracies of the models decreased with increasing risks of 1-year and 5-year deaths (Fig 10). The lines of the accuracies of predicting the risk of 1-year death crossed at 0.3 of the risk of 1-year death (Fig 10A). The accuracies for the risk of 1-year death using the LRM, SVM model, ensemble model, and deep learning model showed similar patterns, and were more than 0.9 at 0.1 of the risk of 1-year death, which was higher than those using the five-cluster model. Moreover, for the prediction of 5-year death, the accuracies of the LRM, SVM model, ensemble model, and deep learning model were higher than that of the five-cluster model (Fig 10B). The accuracies of the deep learning model, LRM, SVM, and ensemble model were almost the same, more than 0.7, at which the interval of 5-year death was about 0.4.

The sensitivities and specificities of the models showed a negative relationship at different cutoff points for clusters (Fig 11). To predict both 1- and 5-year deaths, the five-cluster model

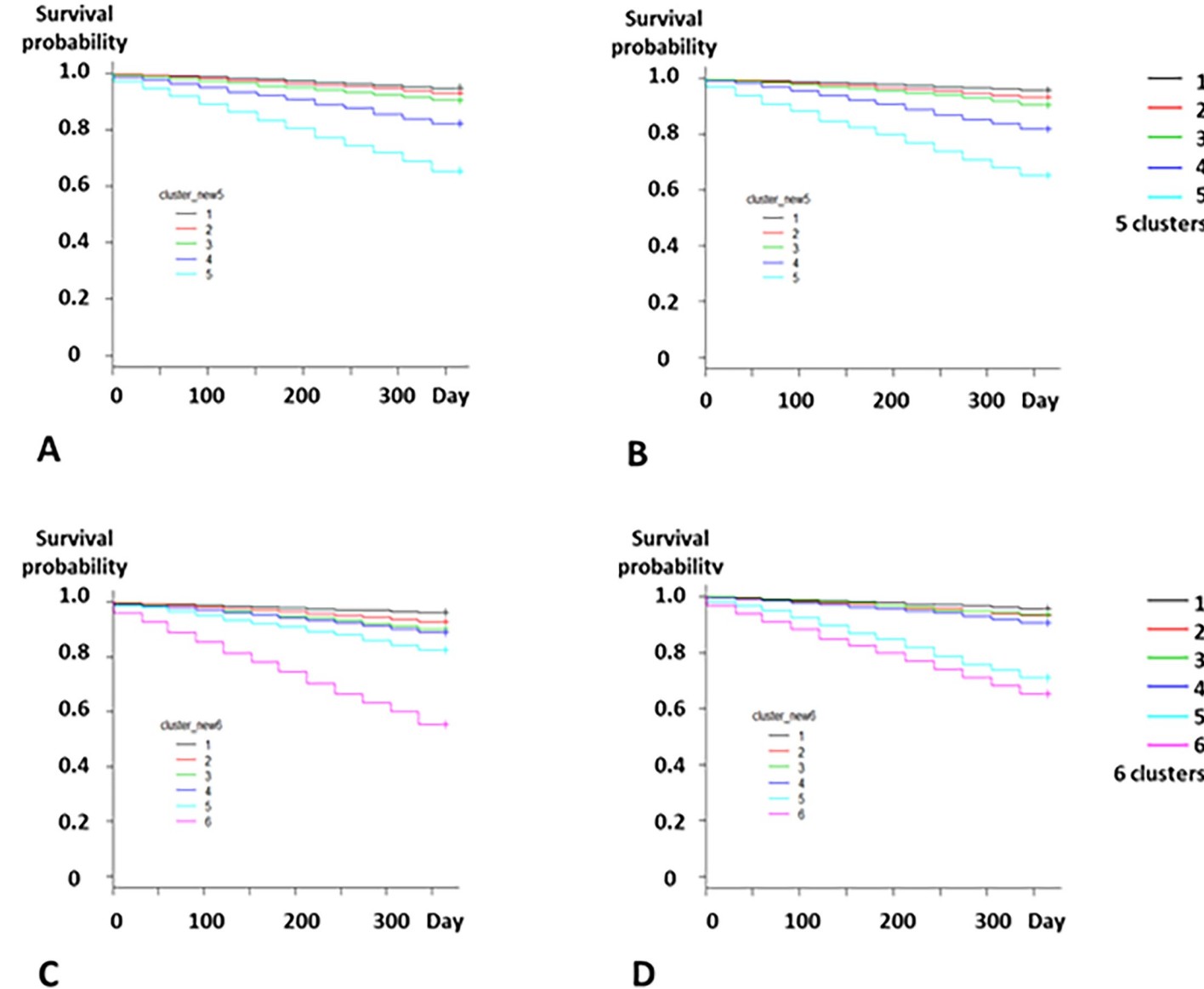

**Fig 7. Association between number of clusters and 1-year mortality.** A. Five clusters in the development dataset. B. Five clusters in the validation dataset. C. Six clusters in the development dataset. D. Six clusters in the validation dataset. The Kaplan-Meier survival curves show that the low-risk group had the highest survival probability in both datasets.

with Cluster 1 as the cutoff point showed higher sensitivities (0.915, 0.860) than the other models. The sensitivities of the LRM, SVM model, ensemble model, deep learning model, and NRI were low. On the other hand, the specificities of the five-cluster model with Cluster 4 as the cutoff point were high (1-year death, 0.829; 5-year death, 0.884), but lower than those of other models (LRM, 0.996, 0.890; SVM model, 0.999, 0.919; ensemble model, 0.999, 0.910; deep learning model, 0.998, 0.855; NRI, 0.937, 0.961, for 1- and 5-year deaths, respectively).

## Total-care system for hemodialysis patients

Considering the characteristics of the machine learning models, for our system, we adopted an ensemble model with the K-means method and SVM for use in clinical settings. Our recommended system is as follows (Fig 12): After clustering, the patients in clusters 1 to 3 are followed

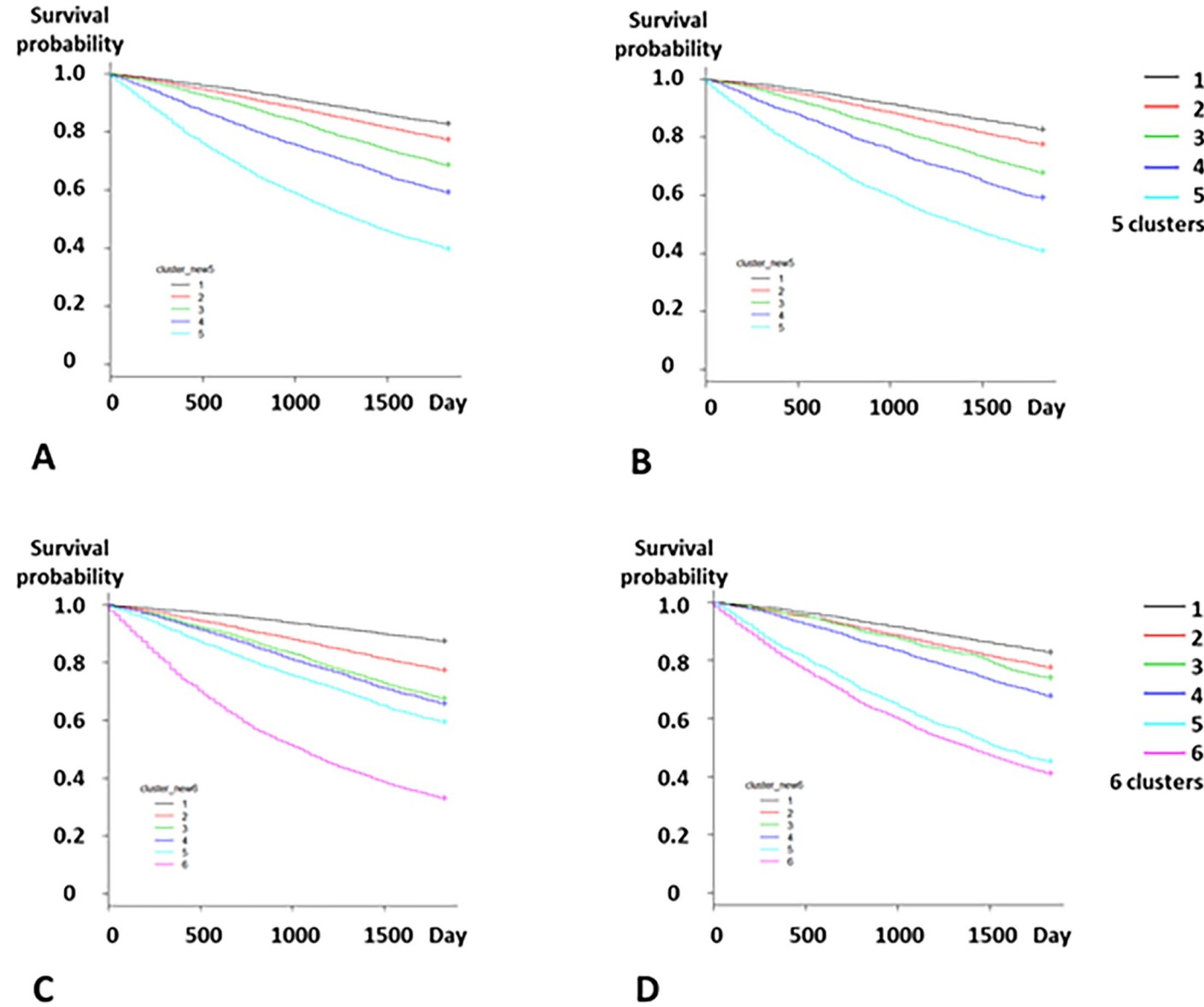

**Fig 8. Association between number of clusters and 5-year mortality.** A. Five clusters in the development dataset. B. Five clusters in the validation dataset. C. Six clusters in the development dataset. D. Six clusters in the validation dataset. Kaplan-Meier survival curves show that the low-risk group had the highest survival probability for the both datasets.

up periodically, because some of them may be classified in Cluster 4 or 5 in the future. Then, the patients in Clusters 4 and 5 are examined using the SVM models. If they are diagnosed to be at a high risk, they undergo detailed medical examinations. If diseases or aggravation of comorbid conditions are diagnosed, intervention and therapy are provided. If not, they are followed up as high-risk patients more frequently and thoroughly than the patients in Clusters 1 to 3.

## Discussion

There are various types of machine learning, whose mechanisms cannot be fully understood by humans, and are called black boxes. Thus, an explainable machine learning model has been studied. Among the types of machine learning, K-means is based on the least square method,

**Table 2. Clusters and risk of 1-year death.**

| A | | |
|---|---|---|
| Cluster number | Development dataset | Validation dataset |
| 1 | Reference | Reference |
| 2 | 1.36 (1.16, 1.6) | 1.58 (1.33, 1.88) |
| 3 | 1.83 (1.58, 2.13) | 2.22 (1.9, 2.61) |
| 4 | 3.51 (2.95, 4.18) | 4.36 (3.64, 5.24) |
| 5 | 7.14 (6.27, 8.14) | 8.86 (7.68, 10.21) |
| B | | |
| Cluster number | Development dataset | Validation dataset |
| 1 | Reference | Reference |
| 2 | 1.87 (1.53, 2.28) | 1.58 (1.32, 1.87) |
| 3 | 2.55 (2.12, 3.08) | 1.51 (1.08, 2.11) |
| 4 | 2.89 (2.39, 3.5) | 2.22 (1.89, 2.6) |
| 5 | 4.62 (3.75, 5.69) | 7.2 (5.95, 8.71) |
| 6 | 12.76 (10.72, 15.18) | 8.79 (7.62, 10.13) |

A. Five clusters.

B. Six clusters.

Values are HRs with 95% CIs of Clusters 2 to 5 compared with Cluster 1.

Abbreviations: HR, hazard ratio; CI, confidence interval.

and is more understandable than other models. Moreover, SVM can be used to predict patients' prognosis. In this work, we developed an explainable ensemble model for the prediction of patients' prognosis, which was composed of K-means and SVM. Hemodialysis patients were categorized into five clusters by the K-means method on their basis of baseline characteristics, which reflected the risk of death. Then, we developed machine learning and statistical

**Table 3. Clusters and risk of 5-year death.**

| A | | |
|---|---|---|
| Cluster number | Development dataset | Validation dataset |
| 1 | Reference | Reference |
| 2 | 1.36 (1.27, 1.45) | 1.34 (1.25, 1.42) |
| 3 | 1.98 (1.87, 2.1) | 2.04 (1.92, 2.16) |
| 4 | 2.79 (2.59, 3.01) | 2.8 (2.59, 3.01) |
| 5 | 5.05 (4.78, 5.33) | 4.9 (4.64, 5.18) |
| B | | |
| Cluster number | Development dataset | Validation dataset |
| 1 | Reference | Reference |
| 2 | 1.92 (1.77, 2.08) | 1.34 (1.25, 1.42) |
| 3 | 2.91 (2.7, 3.13) | 1.57 (1.4, 1.77) |
| 4 | 3.11 (2.88, 3.35) | 2.03 (1.92, 2.15) |
| 5 | 3.92 (3.59, 4.28) | 4.28 (3.94, 4.66) |
| 6 | 8.82 (8.2, 9.49) | 4.9 (4.63, 5.17) |

A. Five clusters.

B. Six clusters.

Values are HRs with 95% CIs of Clusters 2 to 5 compared with the cluster 1.

Abbreviations: HR, hazard ratio; CI, confidence interval.

**Table 4. Baseline characteristics of clusters in validation dataset.**

| | 1 | 2 | 3 | 4 | 5 | p |
|---|---|---|---|---|---|---|
| N | 10358 (25.9) | 8935 (22.4) | 10266 (25.7) | 2660 (6.7) | 7711 (19.3) | |
| Age (years) | 59.4 ± 12.2 | 64.6 ± 11.6 | 64.7 ± 10.5 | 72.8 ± 11.3 | 74 ± 9 | <0.0001 |
| Male (%) | 9755 (94.2) | 692 (7.7) | 7675 (74.8) | 1708 (64.2) | 4717 (61.2) | <0.0001 |
| DM (%) | 29 (0.3) | 696 (7.8) | 9579 (93.3) | 0 (0) | 2680 (34.8) | <0.0001 |
| CGN (%) | 7723 (74.6) | 5391 (60.3) | 28 (0.3) | 0 (0) | 2793 (36.2) | <0.0001 |
| Nephrosclerosis (%) | 1 (0) | 0 (0) | 0 (0) | 2660 (100) | 9 (0.1) | <0.0001 |
| CVD (%) | 1201 (11.6) | 936 (10.5) | 2247 (21.9) | 622 (23.4) | 2462 (31.9) | <0.0001 |
| BMI (kg/m$^2$) | 21.8 ± 3.1 | 19.7 ± 2.8 | 22.8 ± 3.5 | 21.1 ± 3.4 | 20 ± 3.1 | <0.0001 |
| Albumin (g/dL) | 3.9 ± 0.3 | 3.8 ± 0.3 | 3.8 ± 0.3 | 3.7 ± 0.4 | 3.3 ± 0.4 | <0.0001 |
| Sodium (mEq/L) | 139.4 ± 2.9 | 139.4 ± 3.0 | 138.6 ± 3.1 | 139 ± 3.2 | 138.2 ± 3.8 | <0.0001 |
| Potassium (mEq/L) | 5.3 ± 0.7 | 5.2 ± 0.7 | 5.1 ± 0.8 | 4.9 ± 0.8 | 4.4 ± 0.7 | <0.0001 |
| Calcium (mg/dL) | 9.4 ± 0.8 | 9.4 ± 0.8 | 9.1 ± 0.7 | 9.3 ± 0.8 | 9.4 ± 0.8 | <0.0001 |
| Phosphorus (mg/dL) | 5.8 ± 1.3 | 5.3 ± 1.3 | 5.5 ± 1.3 | 5.1 ± 1.3 | 4.3 ± 1.1 | <0.0001 |
| Creatinine (mg/dL) | 13.2 ± 2.3 | 10.2 ± 1.8 | 10.6 ± 2.3 | 10 ± 2.8 | 8.1 ± 2.2 | <0.0001 |
| Total cholesterol (mg/dL) | 146.5 ± 30.5 | 171 ± 34.6 | 152 ± 33.9 | 155.5 ± 34.2 | 146.4 ± 34.1 | <0.0001 |
| CRP (mg/dL) | 0.3 ± 0.9 0.1 (0.1, 0.3) | 0.3 ± 0.9 0.1 (0, 0.2) | 0.3 ± 1.0 0.1 (0.1, 0.3) | 0.6 ± 1.5 0.1 (0.1, 0.4) | 1.2 ± 2.5 0.3 (0.1, 1.1) | <0.0001 |
| Hemoglobin (g/dL) | 10.8 ± 1.2 | 10.4 ± 1.1 | 10.6 ± 1.1 | 10.4 ± 1.2 | 9.9 ± 1.3 | <0.0001 |
| nPCR (g/kg/day) | 0.9 ± 0.2 | 1 ± 0.2 | 0.9 ± 0.2 | 0.9 ± 0.2 | 0.7 ± 0.1 | <0.0001 |
| Vintage (years) | 10.8 ± 7.3 9 (5.1, 14.8) | 11.2 ± 7.6 9.4 (5.3, 15.6) | 4.9 ± 3.3 4.1 (2.4, 6.7) | 5.5 ± 4.1 4.3 (2.6, 7.2) | 7.1 ± 6.2 5.1 (2.8, 9.3) | <0.0001 |
| Kt/V | 1.4 ± 0.2 | 1.7 ± 0.3 | 1.3 ± 0.2 | 1.4 ± 0.3 | 1.4 ± 0.3 | <0.0001 |
| Ultrafiltration (%) | 4.5 ± 1.5 | 4.9 ± 1.7 | 4.4 ± 1.6 | 4.2 ± 1.8 | 3.6 ± 2.2 | <0.0001 |
| NRI | | | | | | <0.0001 |
| Low risk (%) | 9220 (89.0) | 6531 (73.1) | 8321 (81.1) | 1812 (68.1) | 2820 (36.6) | |
| Medium risk (%) | 1003 (9.7) | 1927 (21.6) | 1713 (16.7) | 597 (22.4) | 2839 (36.8) | |
| High risk (%) | 135 (1.3) | 477 (5.3) | 232 (2.3) | 251 (9.4) | 2052 (26.6) | |

Variables are expressed as mean±standard deviation. Vintage and CRP are also shown as median and interquartile range. Intergroup comparisons of parameters were performed using the chi-square test, t-test, and the Mann-Whitney U test as appropriate.

Abbreviations: DM, diabetes mellitus as a cause of end-stage renal disease; CGN, chronic glomerulonephritis; CVD, cardiovascular disease; BMI, body mass index; CRP, C-reactive protein; nPCR, normalized protein catabolic rate; NRI, nutritional risk index.

models, and compared their performances. The ensemble model of the K-means method and SVM showed the highest accuracy of the prediction of death. Although some studies showed a high accuracy of the prediction of dialysis patients' death using machine learning models, the

**Table 5. Number of endpoints in validation dataset.**

| | 1 | 2 | 3 | 4 | 5 | p |
|---|---|---|---|---|---|---|
| N | 10358 | 8935 | 10266 | 2660 | 7711 | |
| 1-year death (%) | 221 (2.1) | 299 (3.3) | 482 (4.7) | 240 (9) | 1343 (17.4) | <0.0001 |
| 5-year death (%) | 1784 (17.2) | 1993 (22.3) | 3299 (32.1) | 1084 (40.8) | 4541 (58.9) | <0.0001 |
| Cause of 5-year death | | | | | | |
| CVD caused death (%) | 487 (4.7) | 547 (6.1) | 875 (8.5) | 297 (11.2) | 1354 (17.6) | <0.0001 |
| Infection-caused death (%) | 94 (0.9) | 91 (1.0) | 130 (1.3) | 72 (2.7) | 362 (4.7) | <0.0001 |
| Other-cause death (%) | 1203 (11.6) | 1355 (15.2) | 2294 (22.3) | 715 (26.9) | 2825 (36.6) | <0.0001 |

The values are number of deaths (%).

Abbreviations: CVD, cardiovascular disease.

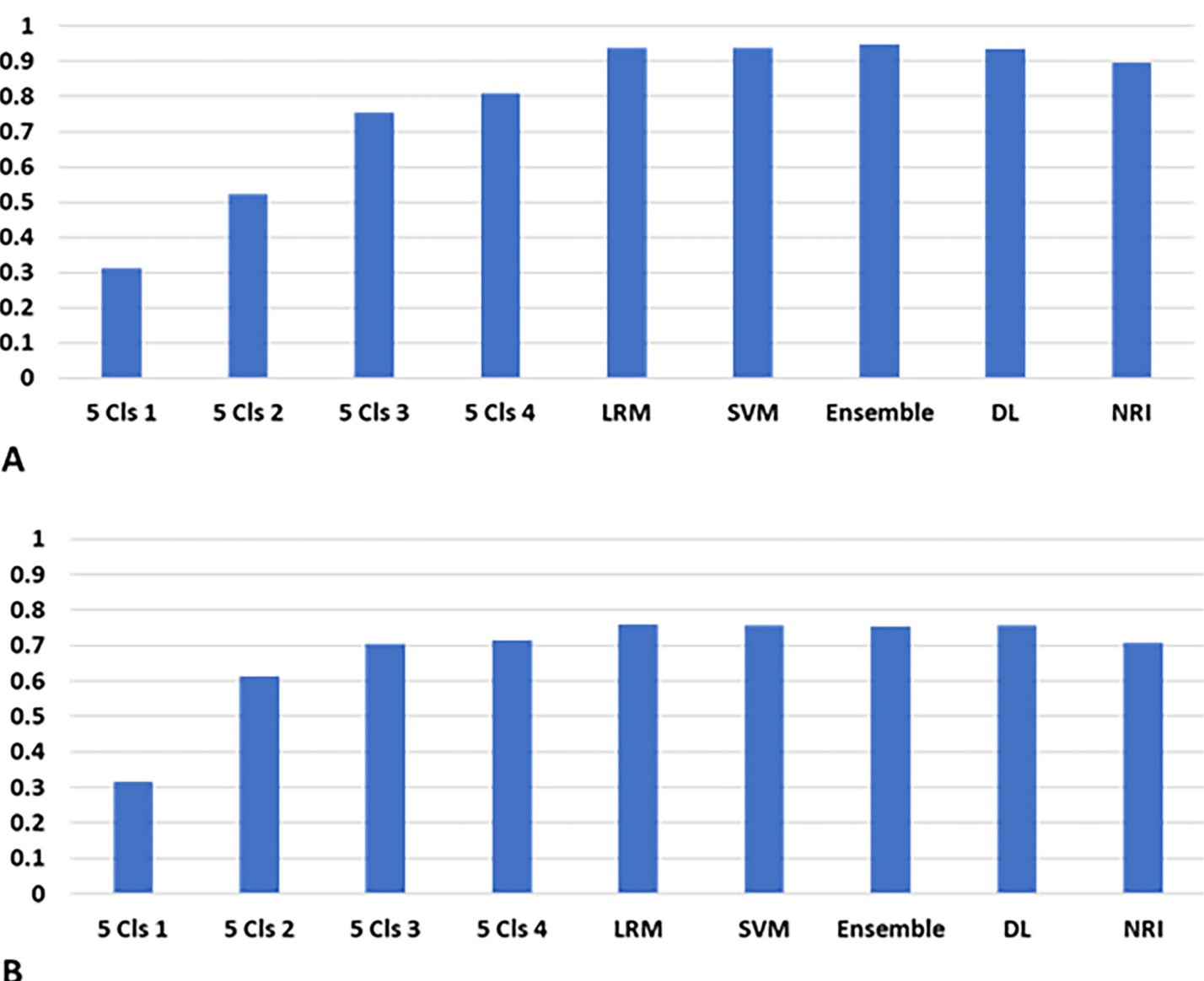

**Fig 9. Accuracies of predicting 1-year and 5-year deaths.** A. 1-year death. B. 5-year death. The accuracies of the models were evaluated using the validation dataset. Abbreviations: 5 Cls 1, the cutoff point was the first cluster of Cluster 5; LRM, multivariate logistic regression model; SVM, support vector machine; Ensemble, ensemble model of K-means method and SVM models; DL, deep learning model; NRI, nutritional risk index.

internal structures of the models were difficult to understand [11–13]. There is a tradeoff relationship between the accuracy of prediction and the transparency of algorithms [14]. We attempted to achieve a balance by developing a blended system, which we found useful for identifying patients at a high risk of death, and which was easily applicable to clinical settings.

The International Society of Renal Nutrition and Metabolism proposed an algorithm for the nutritional management and support of chronic kidney disease patients [15]. In the algorithm, multiple nutritional examinations, such as measurement of dietary nutritional intakes, subjective global assessment, and anthropometrics, are recommended [15]. However, it is difficult for all of these nutritional examination results to be digitized and evaluated by machine learning models. Moreover, a systematic review of the studies of the data-driven population segmentation analysis pointed out that a perfect diagnosis is not always guaranteed; and the

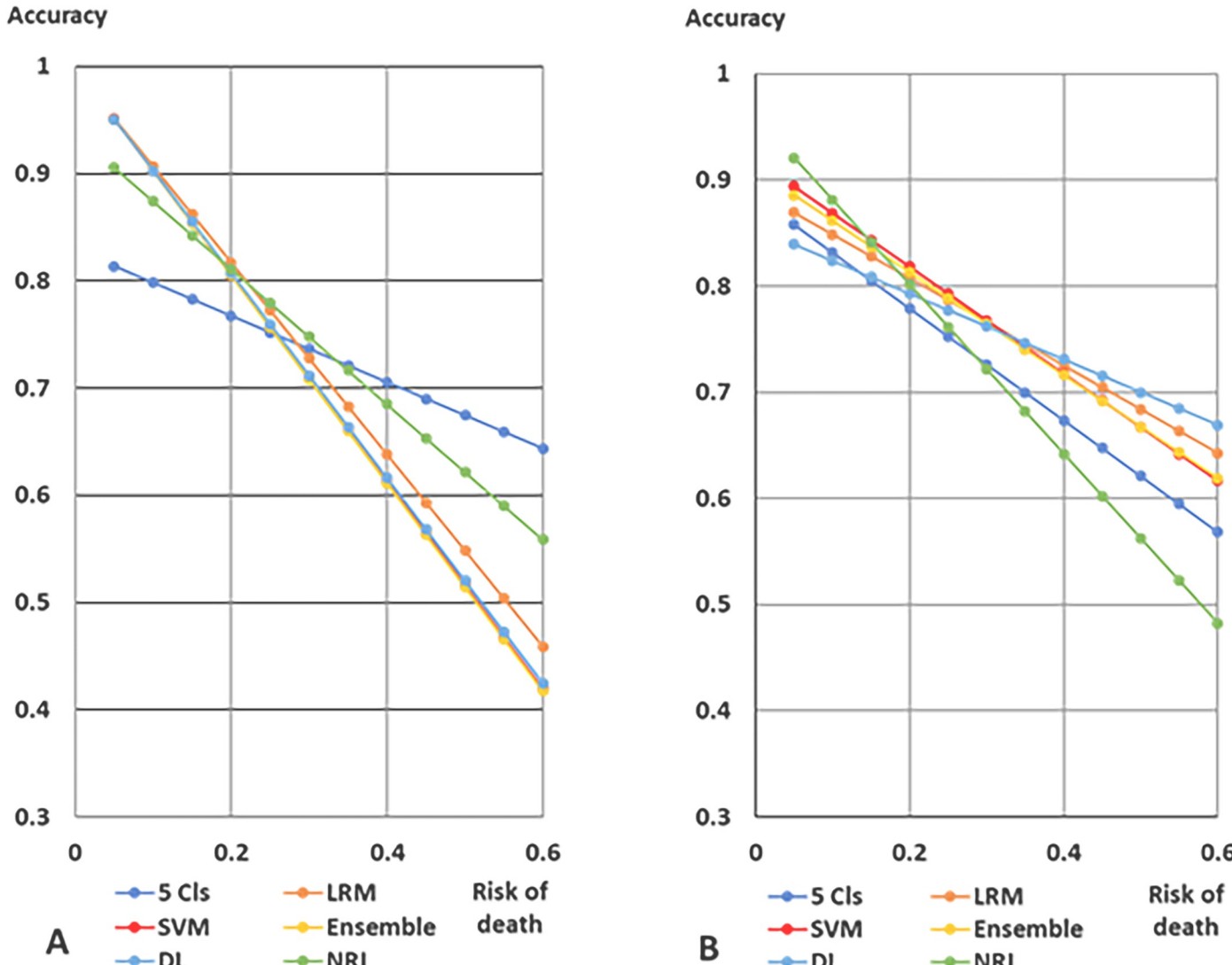

**Fig 10. Accuracies and risk of 1-year and 5-year deaths.** A. 1-year death. B. 5-year death. The accuracies of the model were calculated at 1year and 5-year death. Abbreviations: 5 Cls, the cutoff point was the fourth cluster of clusters; LRM, multivariate logistic regression model; SVM, support vector machine; ensemble, ensemble model of K-means method and SVM models; DL, deep learning model; NRI, nutritional risk index.

review suggested the importance of assessing the segmentation outcome with a combination of statistical reasoning, clinical judgement, and policy implication [16]. Therefore, we did not leave the entire diagnosis to be performed by a machine learning system, and instead developed the ensemble model as part of the medical system. The ensemble model and detailed medical examinations can complement each other, which enhances the robustness of this system.

According to the JSDT annual report in 2015, the mean age of Japanese dialysis patients was 67.86 years, 64.3% were male, and the causes of ESKD were DM (38.4%), CGN (29.8%), nephrosclerosis (9.5%) [1]. Considering these basic statistics, our system could divide the patients into the five clusters reflecting their baseline characteristics (Table 6). These characteristics were risk factors for death in their prognosis [3, 4, 6]. For example, the risks of all-cause death, CVD- and infection-caused deaths in Cluster 5 were higher than those in other clusters.

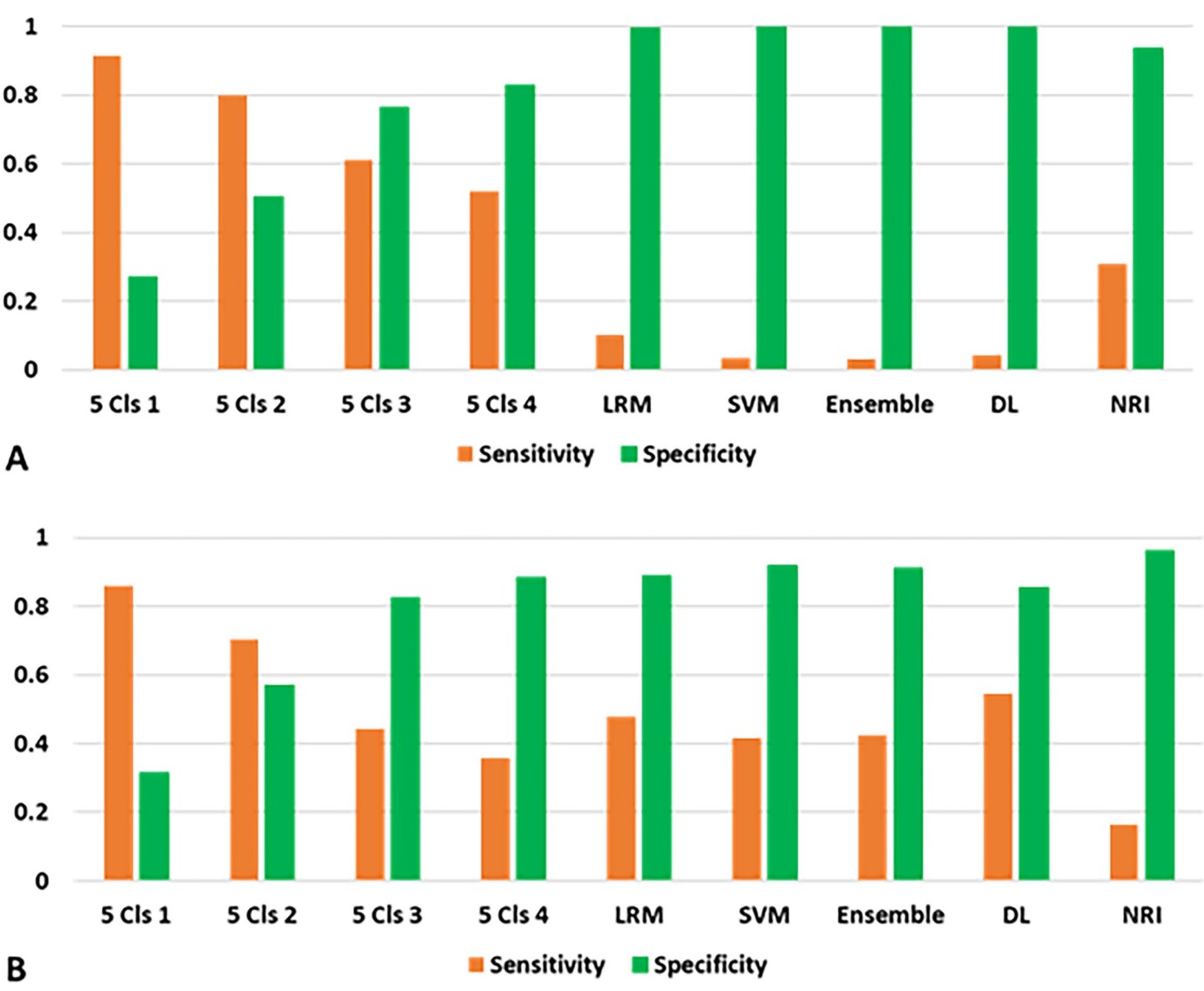

**Fig 11. Sensitivities and specificities of models to predict risks of 1-year and 5-year deaths.** A. 1-year death. B. 5-year death. Abbreviations: 5 Cls 1, the cutoff point was the first cluster of five clusters; LRM, multivariate logistic regression model; SVM, support vector machine; ensemble, ensemble model of K-means method and SVM models; DL, deep learning model; NRI, nutritional risk index.

And Cluster 5 showed lower serum albumin and creatinine levels and lower nPCR, which are nutritional factors, than the other clusters, and included a large number of patients with high and medium risks of NRI of 26.6% and 36.8%, respectively. Moreover, a high serum CRP level, which indicates inflammation, was also observed in Cluster 5. Inflammation is often observed in ESKD patients with malnutrition, and this complex state of malnutrition and inflammation is called protein energy wasting (PEW) [5]. PEW causes CVD which is a risk factor for death [5, 8]. The classification of an elderly patient with PEW into Cluster 5 indicates that the treatment of PEW should be of the highest priority.

Our system could clearly distinguish patients with DM (Cluster 3) or nephrosclerosis (Cluster 4) from those with other conditions. The patients in Cluster 4 showed a higher risk of death than those in Cluster 3. What factor made this difference? Both DM and aging are the main causes of CVD in hemodialysis patients [17]. According to a systematic review, they are risk

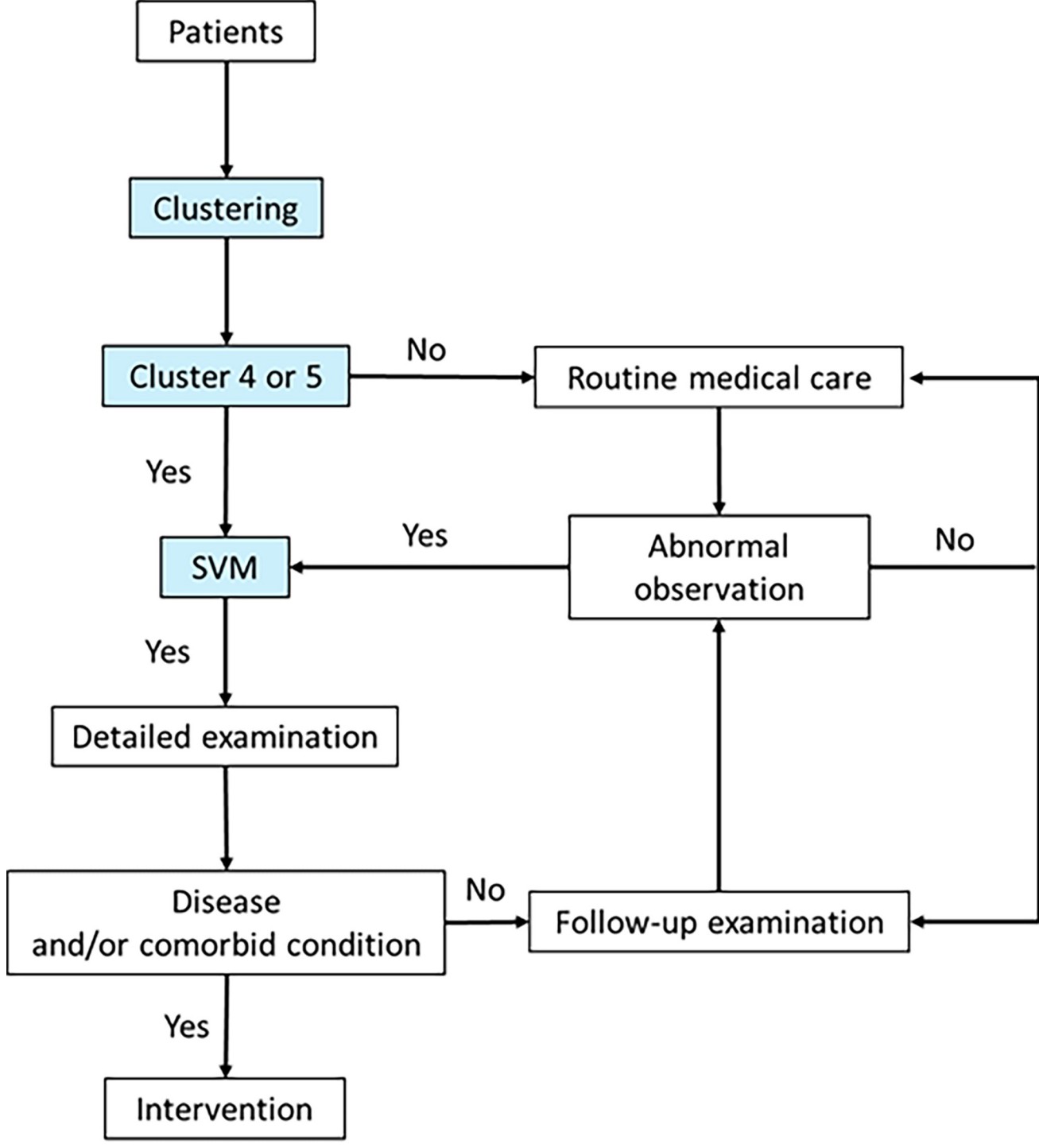

**Fig 12. Diagnostic system using ensemble model.** Our system is as follows: a. Extract high-risk patients (Clusters 4 and 5) on the basis of sensitivity, specificity, and characteristics of the clusters. b. Examine in detail patients in Clusters 4 and 5 using SVM models, and routinely follow up depending on their risk of death. The blue part can be examined using the ensemble model. Abbreviation: SVM, support vector machine models.

factors for all-cause and CVD-caused deaths [18]. In our study, no clear differences were observed in the other risk factors reported in the systematic review, such as history of CVD,

**Table 6. Specific characteristics of clusters.**

| Cluster number | Specific characteristics |
| --- | --- |
| 1 | Young, male, CGN |
| 2 | Female, CGN |
| 3 | DM |
| 4 | Elderly, nephrosclerosis |
| 5 | Elderly, CVD, malnutrition, inflammation |

Abbreviations: CGN, chronic glomerulonephritis as a cause of end-stage renal disease; DM, diabetes mellitus as a cause of end-stage renal disease; CVD, cardiovascular disease.

BMI, hemoglobin level, serum albumin, and CRP levels, between Clusters 3 and 4 [18]. The only factors different between these clusters were the causes of ESKD and age; patients in Cluster 4 were about 8 years older than those in Cluster 3. DOPPS showed no statistically significant difference in mortality rate between patients with DM and hypertension [19]. It is possible that age itself might have caused the survival difference. DM has been the leading cause of ESKD in Japan, and the number of dialysis patients with DM has been stable over the past few years [1]. In contrast, nephrosclerosis is caused by aging and hypertension, and the number of patients with nephrosclerosis has been increasing with the aging of the population in Japan [1]. Elderly patients with nephrosclerosis should be paid more attention, because they are at a high risk of death, and will be a majority among dialysis patients in the near future.

Similar to our study, a cohort study of the health care system in Singapore showed a relationship among K-means clusters, healthcare utilization pattern, and mortality [20]. Why do the clusters obtained by the K-means method reflect the patients' prognosis in the Singapore study and our study? The cluster centers were obtained using Eq (5). $\mu_j$ is a vector equal to the mean of all data of patients in Cluster $j$. That is, patients in Cluster $j$ are distributed in an m-dimensional sphere with the center at $\mu_j$. In this study, the number of clusters was determined by the links with the risk of death as an important true endpoint, which showed that $\mu_j$ was strongly associated with risk of death. On the basis of these theoretical backgrounds, each cluster had specific characteristics of risk factors for death, such as gender, causes of ESKD, and PEW (Table 6). In the risk prediction models using standard statistics, the variables are often arbitrarily selected, whereas in machine learning, patients' features are extracted from their numerical data, even though a human does not provide sufficient information. There is a possibility that this feature extraction can clarify the new pathophysiological characteristics of diseases. For example, the five clusters in this study, which had different numerical features, may have different courses of change in their body condition after dialysis initiation. Thus, new unknown research seeds will be mined by machine learning.

The performance of machine learning is often evaluated by the accuracy of classification. When using the validation data, the ensemble model showed a higher accuracy of the prediction of death than other models. The analysis of machine learning models, e.g., SVM and deep learning models, is a black box [21]. Because our ensemble model was composed of the K-means method and SVM model, this combined system of classification and prediction made the results interpretable with high accuracy, and closely matched the clinical decision-making process. The practical applications of this kind of machine learning model have never been reported.

Because accuracy is determined by the incident number of events, it changes with the composition of the sample population. Thus, we evaluated the changes in the accuracies of the models with the changes in the risk of death. The risks of 1-year death in Japan and USA are

9.6% and 13.4%, and those of 5-year death in Japan, Italy, and USA are 39.5%, 44.4%, and 58%, respectively [1, 2, 22, 23]. In the simulation using these populations, the machine learning models could show high accuracies, and effectively predicted the prognosis of ESKD patients.

The classification performance of diagnostic tests is commonly evaluated in terms of sensitivity and specificity. The machine learning models in this study showed their high specificity to predict 1-year and 5-year deaths. High specificity means that the models have a small number of false-positive patients. That is, when a patient is diagnosed to be positive for a risk by the models, the possibility of the presence of a disease is high. Therefore, it could be said that the diagnosis obtained using the models with high specificities is useful to confirm the diagnosis. On the other hand, because the sensitivities of SVM and deep learning models were low, they were not appropriate for screening high-risk patients. The sensitivities of the K-means method using clusters were higher than those of the other models. The clusters might be useful for identifying the high-risk patients.

Our system is applicable to clinical settings in the context of its limitations. First, in this study, JRDR data were used. This data were obtained from 98.8% of dialysis patients in Japan, reflecting the real-world of dialysis patients in Japan. Because our system was developed using these data, its accuracy for Japanese or Asian patients is high, but the results using data from other countries might be biased by the sampling of patients. Second, we did not include patients with missing data in this study, which might cause a selection bias. Third, the JRDR data did not include sufficient data for assessing malnutrition, blood pressure, comorbid conditions, and medications. And, we were unable to evaluate the effects of the differences in the baseline characteristics such as dietary intake; comorbid conditions such as DM and hypertension; and medications such as hypoglycemic and antihypertensive medicines on the clustering. Further studies are needed to evaluate the relationship between these factors and clustering. Thus, such data would improve the accuracy of the models.

## Conclusions

We developed a novel system using machine learning algorithms that analyzes hemodialysis patients' data, categorizes the patients on the basis of their characteristics, and identifies patients at a high risk of death. The new approach has a strong potential to guide treatments and improve hemodialysis patients' prognosis.

## Acknowledgments

The data reported here have been provided by JSDT. The interpretation and reporting of these data are the responsibility of the authors and in no way seen as an official policy or interpretation of the JSDT.

## Author Contributions

**Conceptualization:** Eiichiro Kanda, Bogdan I. Epureanu, Yuki Tsuruta, Kan Kikuchi, Masanori Abe, Ikuto Masakane, Kosaku Nitta.

**Data curation:** Eiichiro Kanda, Bogdan I. Epureanu, Yuki Tsuruta, Kan Kikuchi, Ikuto Masakane, Kosaku Nitta.

**Formal analysis:** Eiichiro Kanda, Bogdan I. Epureanu, Taiji Adachi.

**Investigation:** Eiichiro Kanda, Bogdan I. Epureanu, Yuki Tsuruta, Kan Kikuchi, Masanori Abe.

**Methodology:** Eiichiro Kanda, Bogdan I. Epureanu, Taiji Adachi, Yuki Tsuruta, Kan Kikuchi.

**Project administration:** Eiichiro Kanda, Naoki Kashihara, Kosaku Nitta.

**Resources:** Eiichiro Kanda, Kosaku Nitta.

**Software:** Eiichiro Kanda.

**Supervision:** Naoki Kashihara, Masanori Abe, Kosaku Nitta.

**Validation:** Taiji Adachi, Naoki Kashihara, Masanori Abe, Ikuto Masakane.

**Visualization:** Eiichiro Kanda.

**Writing – original draft:** Eiichiro Kanda.

**Writing – review & editing:** Bogdan I. Epureanu, Taiji Adachi, Yuki Tsuruta, Kan Kikuchi, Naoki Kashihara, Masanori Abe, Ikuto Masakane, Kosaku Nitta.

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
