## [Decision Letter · Decision Letter 0]

22 Apr 2020

PONE-D-20-09660

Application of explainable ensemble artificial intelligence model to categorization of hemodialysis-patient and treatment using nationwide-real-world data in Japan

PLOS ONE

Dear Professor Kanda,

Thank you for submitting your manuscript to PLOS ONE. After careful consideration, we feel that it has merit but does not fully meet PLOS ONE’s publication criteria as it currently stands. Therefore, we invite you to submit a revised version of the manuscript that addresses the points raised during the review process.

The system the authors proposed is promising, but there is some questions remained to clarify the importance. Please read the reviewer's comments and let us know your opinion.

In addition, the journal staff provided the comment as follows.

Whether it meets PLOS ONE criteria for papers that describe new methods or software for applications? Specifically these reports must meet the criteria of utility validation and availability which are described in detail at http://journals.plos.org/plosone/s/submission-guidelines#loc-methods-software-databases-and-tools. 

Please include the reply to the comment when you send the revised manuscript.

We would appreciate receiving your revised manuscript by Jun 06 2020 11:59PM. To enhance the reproducibility of your results, we recommend that if applicable you deposit your laboratory protocols in protocols.io, where a protocol can be assigned its own identifier (DOI) such that it can be cited independently in the future. For instructions see: http://journals.plos.org/plosone/s/submission-guidelines#loc-laboratory-protocols

We look forward to receiving your revised manuscript.

Kind regards,

Kojiro Nagai

Academic Editor

PLOS ONE

Journal Requirements:

'The funders had no role in study design, data collection and analysis, decision to

publish, or preparation of the manuscript.'

Additional Editor Comments (if provided):

Reviewers' comments:

Reviewer's Responses to Questions

**Comments to the Author**

1. Is the manuscript technically sound, and do the data support the conclusions?

Reviewer #1: Yes

Reviewer #2: Yes

2. Has the statistical analysis been performed appropriately and rigorously? 

Reviewer #1: Yes

Reviewer #2: Yes

3. Have the authors made all data underlying the findings in their manuscript fully available?

Reviewer #1: Yes

Reviewer #2: Yes

4. Is the manuscript presented in an intelligible fashion and written in standard English?

Reviewer #1: Yes

Reviewer #2: Yes

5. Review Comments to the Author

Reviewer #1: General comments

This manuscript described the construction and evaluation of artificial intelligence model to categorization of hemodialysis-patient for survival rate using nationwide-real-world data in Japan. This study contains some novel factors. However, there are several concerns that should be addressed.

Comments

1. This clustering system might be useful for identifying the high-risk patients. However, does the simplified clustering system (specific characteristics of clusters) reflect the results derived by machine learning?

2. If we want a prognostic prediction model for an individual patient, can we make a prognostic prediction by clustering information from the patient in front of us? I would also like to see a clear distinction between the scoring system and the clustering system.

Reviewer #2: The manuscript by Kanda E et al. investigated the characteristics of hemodialysis patients using machine learning model, and its usefulness for screening hemodialysis patients at a high risk of one-year death using the nation-wide database of the Japanese Society for Dialysis Therapy (JSDT), and found that the five clusters clearly distinguished the groups on the basis of their characteristics and reflected their prognosis. The paper is well written and the topic is important for analyzing the present and future situations of hemodialysis patients in Japan. I have some comments as follows:

1. The mean ages of clusters 4 and 5 were older and the C-reactive protein levels of them higher than those of other groups. Therefore, the authors should discuss the cause of death because it seems to be different in the five clusters.

2. Did the dataset of the JSDT include data on blood pressure and medication use? If not, please describe in the discussion. If included, please explain why the authors did not use them as the baseline characteristics in their model.

3. Minor: page 7, line 13; albuminlevel should be albumin level.

6. PLOS authors have the option to publish the peer review history of their article (what does this mean?). If published, this will include your full peer review and any attached files.

Reviewer #1: No

Reviewer #2: No

---

## [Author Response · Author response to Decision Letter 0]

28 Apr 2020

PLOS ONE 

Academic Editor

Kojiro Nagai

Dear Dr. Nagai:

Thank you very much for your letter with the reviewers’ comments and for your helpful remarks on our paper. We have revised the manuscript in accordance with the reviewers’ comments and suggestions. The changes are written with tracked changes. This cover letter includes our point-by-point responses to the reviewers’ comments, which we hope to have been addressed satisfactorily.

Reviewer #1: General comments

This manuscript described the construction and evaluation of artificial intelligence model to categorization of hemodialysis-patient for survival rate using nationwide-real-world data in Japan. This study contains some novel factors. However, there are several concerns that should be addressed.

>> Thank you very much for your comments.

Comments

1. This clustering system might be useful for identifying the high-risk patients. However, does the simplified clustering system (specific characteristics of clusters) reflect the results derived by machine learning?

>> Thank you very much for your question. There are various types of machine learning, whose mechanisms cannot be fully understood by humans, and are called black boxes. Thus, an explainable machine learning model has been studied. K-means is based on the least square method, and is more understandable than other models. Moreover, the support vector machine (SVM) can be used to accurately predict patients’ prognosis. In this work, we developed an explainable ensemble model for the prediction of patients’ prognosis, which is composed of K-means and SVM. Thus, because K-means (clustering system) is part of the ensemble model, the clusters showed intermediate results of the ensemble model. Considering the difficulty for readers to understand the models, the above explanation was provided in Discussion: Page 28, paragraph 1. 

2-1. If we want a prognostic prediction model for an individual patient, can we make a prognostic prediction by clustering information from the patient in front of us? 

>> Considering your question No. 1, because the clustering was part of the ensemble model, to evaluate accurately patients’ prognosis, not only the clustering but also analyses by SVM are necessary. The trained ensemble model can be easily applicable to other patients in clinical settings. 

2-2. I would also like to see a clear distinction between the scoring system and the clustering system.

>> In this study, because clustering is a part of the analysis of the ensemble model, the explanation of the difference between the risk scoring system and machine learning is appropriate for this question.

 The standard scoring system is usually developed using a logistic regression model or a Cox proportional hazards model. These models are constructed on the basis of statistical assumptions. For example, there is an ideal population, and are restrictions of variables, such as the number of variables included in a model, the distribution pattern of error, proportional hazards, and so forth. Moreover, the variables are often used in a very simple linear model, such as β1x1 + β2x2 + + βnxn. Therefore, scoring systems have many restrictions of statistical assumptions, and have a limit of accuracy of prediction.

 On the other hand, in machine learning, a population is not assumed; there is no assumption of the models, and no limit of the number of the variables in the models. Moreover, machine learning can be used to construct nonlinear models. Therefore, machine learning models such as SVM and deep learning can show higher prediction accuracy than scoring systems. In this study, the ensemble model could attain high prediction accuracy because of the combination of K-means and SVM. Therefore, machine learning has less restrictions of model development, and is expected to attain higher prediction accuracy than scoring system.

Moreover, in the scoring system, the variables are arbitrarily selected, whereas in machine learning, patients’ features are extracted from their numerical data, even though a human does not teach anything. There is a possibility that this feature extraction can clarify the new pathophysiological characteristics of diseases. For example, the five clusters in this study, which had different numerical features, may have different courses of change in their body condition after dialysis initiation. Thus, new unknown research seeds will be mined by machine learning. This is described in Discussion: Page 32, line 9.

Reviewer #2: The manuscript by Kanda E et al. investigated the characteristics of hemodialysis patients using machine learning model, and its usefulness for screening hemodialysis patients at a high risk of one-year death using the nation-wide database of the Japanese Society for Dialysis Therapy (JSDT), and found that the five clusters clearly distinguished the groups on the basis of their characteristics and reflected their prognosis. The paper is well written and the topic is important for analyzing the present and future situations of hemodialysis patients in Japan. I have some comments as follows:

>> Thank you very much for your comments.

1. The mean ages of clusters 4 and 5 were older and the C-reactive protein levels of them higher than those of other groups. Therefore, the authors should discuss the cause of death because it seems to be different in the five clusters.

>> Thank you very much for pointing this out. The details of deaths are summarized in Table 5. The percentages of CVD- and infection-caused deaths in Cluster 5 were higher than those in other clusters. These results are described in Results and Discussion: Page 23, Table 5; Page 23, Paragraph 2; Page 30, Paragraph 1.

2. Did the dataset of the JSDT include data on blood pressure and medication use? If not, please describe in the discussion. If included, please explain why the authors did not use them as the baseline characteristics in their model.

>> Blood pressure and medication were important factors for analyzing the characteristics of patients on the basis of the clusters. However, these factors were not included in our dataset. This is described in Discussion as a limitation: Page 34, Paragraph 2, line 7.

3. Minor: page 7, line 13; albuminlevel should be albumin level.

>> Thank you very much.

---

## [Decision Letter · Decision Letter 1]

7 May 2020

Application of explainable ensemble artificial intelligence model to categorization of hemodialysis-patient and treatment using nationwide-real-world data in Japan

PONE-D-20-09660R1

Dear Dr. Kanda,

We are pleased to inform you that your manuscript has been judged scientifically suitable for publication and will be formally accepted for publication once it complies with all outstanding technical requirements.

With kind regards,

Kojiro Nagai

Academic Editor

PLOS ONE

Additional Editor Comments (optional):

Reviewers' comments:

Reviewer's Responses to Questions

**Comments to the Author**

1. If the authors have adequately addressed your comments raised in a previous round of review and you feel that this manuscript is now acceptable for publication, you may indicate that here to bypass the “Comments to the Author” section, enter your conflict of interest statement in the “Confidential to Editor” section, and submit your "Accept" recommendation.

Reviewer #1: All comments have been addressed

Reviewer #2: All comments have been addressed

2. Is the manuscript technically sound, and do the data support the conclusions?

Reviewer #1: Yes

Reviewer #2: Yes

3. Has the statistical analysis been performed appropriately and rigorously? 

Reviewer #1: Yes

Reviewer #2: Yes

4. Have the authors made all data underlying the findings in their manuscript fully available?

Reviewer #1: Yes

Reviewer #2: Yes

5. Is the manuscript presented in an intelligible fashion and written in standard English?

Reviewer #1: Yes

Reviewer #2: Yes

6. Review Comments to the Author

Reviewer #1: In this manuscript, the authors revised their manuscript in accordance with our review.

This manuscript fulfilled our suggestion.

Reviewer #2: The revised manuscript by Kanda E., et al. responded well to the points raised. I have no further critique.

7. PLOS authors have the option to publish the peer review history of their article (what does this mean?). If published, this will include your full peer review and any attached files.

Reviewer #1: No

Reviewer #2: No

---

## [Editor Report · Acceptance letter]

15 May 2020

PONE-D-20-09660R1 

Application of explainable ensemble artificial intelligence model to categorization of hemodialysis-patient and treatment using nationwide-real-world data in Japan 

Dear Dr. Kanda:

I am pleased to inform you that your manuscript has been deemed suitable for publication in PLOS ONE. Congratulations! Your manuscript is now with our production department. 

With kind regards,

on behalf of

Dr. Kojiro Nagai 

Academic Editor

PLOS ONE